# A Near-Field Imaging Method Based on the Near-Field Distance for an Aperture Synthesis Radiometer

Yuanchao Wu, Yinan Li *, Guangnan Song, Haofeng Dou, Dandan Wen, Pengfei Li, Xiaojiao Yang, Rongchuan Lv and Hao Li

China Academy of Space Technology (Xi'an), Xi'an 710100, China; d201477479@hust.edu.cn (Y.W.); songgn@cast504.com (G.S.); haofeng_dou@hust.edu.cn (H.D.); wendd@cast504.com (D.W.); lipf@cast504.com (P.L.); yangxj76@cast504.com (X.Y.); lvrc@cast504.com (R.L.); lih19@cast504.com (H.L.)
* Correspondence: liyn@cast504.com

**Abstract:** For an aperture synthesis radiometer (ASR), the visibility and the modified brightness temperature (BT) are related to the Fourier transform when the distance between the system and the source is in the far-field region. BT reconstruction can be achieved using G-matrix imaging. However, for ASRs with large array sizes, the far-field condition is not satisfied when performing performance tests in an anechoic chamber due to size limitations. Using far-field imaging methods in near-field conditions can introduce errors in the images and fail to correctly reconstruct the BT. Most of the existing methods deal with visibilities, converting near-field visibilities to far-field visibilities, which are suitable for point sources but not good for extended source correction. In this paper, two near-field imaging methods are proposed based on the near-field distance. These methods enable BT reconstruction in near-field conditions by generating improved resolving matrices: the near-field G-matrix and the F-matrix. These methods do not change the visibility measurements and can effectively image both the point source and the extended source in the near field. Simulations of point sources and extended sources in near-field conditions demonstrate the effectiveness of both methods, with F-matrix imaging outperforming near-field G-matrix imaging. The feasibility of both near-field imaging methods is further validated by carrying out experiments on a 10-element Y-array system.

**Keywords:** near-field imaging; ASR; visibility; G-matrix; far-field imaging

## 1. Introduction

In the interest of improving spatial resolution, ASRs have been widely developed in earth remote sensing. Different from the real aperture radiometer, which adopts large antenna mechanical or electrical scanning to image the BT of the scene in the FOV (field of view), the ASR observes the scene through the antenna array [1]. It carries out cross-correlation processing after filtering, amplifying, mixing, and A/D transformation of the received signal, to obtain the visibility of different antenna pairs. In the far-field condition, the reconstructed BT in the FOV can be obtained using the inverse Fourier transform of the visibility. Several ASRs have been studied, such as ESTAR [2], GeoSTAR [3], and MIRAS [4,5]. As the most representative ASR, MIRAS was developed and launched by ESA in 2009 and has been operating ever since.

In earth remote sensing applications, ASRs typically have an extensive array size to meet the high spatial resolution required. For example, MIRAS is equipped with a Y-array with a diameter of 8 m [6]. When these ASRs carry out performance tests and imaging tests in the ground anechoic chambers, the source is usually located in the near field of the array due to site size limitations. If the Fourier transform is used to reconstruct BT, near-field errors will be introduced, resulting in image distortion and inaccurate performance estimation. In addition, with the development of the ASR, it can also be used in security [7], geological exploration [8], all-weather reconnaissance, and

surveillance. For those applications, the target is located in the near-field region, and the curvature of the wave front cannot be neglected. The near-field error is different from the receiver channel errors and baseline errors which can be corrected using the internal calibration method [9,10]. It requires a special near-field error correction method.

Some solutions have been proposed to make the ASR show good imaging performance in near-field conditions. In [11], spherical and circular array architectures are proposed to minimize near-field distortions. This arrangement produces an equivalent far-field volume near its center, so the Fourier transform is still applicable to the BT reconstruction. However, this method only works when the target is located on the focus point. Once the target area is large, this method is not applicable. In [12,13], near-field imaging is simply achieved by subtracting the phase of the near field and adding the phase of the far field. It can be used for the near-field imaging of point sources. For the extended source, this method can simply correct the pixel at boresight and large errors remain in off-boresight areas. In [14], the phase of the near-field visibilities of the scene subtracts the phase of the near-field visibilities of the reference point source to determine the conversion from near-field visibility to far-field visibility. The position of the source element must be close to the reference point source, otherwise the correction effect will be degraded. In [15], the near-field beamforming method is applied to determine the near-field imaging of ASR, and the imaging quality relies on the optimization of the weight vector. [16] adopts the angular spectrum method to ASR near-field imaging. After multi-step processing, the phase of the near-field visibilities is compensated to reconstruct BT. Although this method can be used for near-field imaging of both point source and extended source, it increases the complexity of imaging processing.

This paper presents the derivation of the visibility formula under the far-field condition and the corresponding G-matrix imaging method. By analyzing the visibility under the near-field condition, two near-field imaging methods based on the near-field distance are proposed. Simulations and experimental results show that the two methods are feasible for near-field imaging and are effective for both point sources and extended sources.

## 2. Far-Field Imaging

As illustrated in Figure 1, the array of the ASR is located entirely in the plane z = 0, and the target source lies on the plane z = h. For two antenna elements $P_i$ and $P_j$ in the array, their coordinates are $(x_i, y_i, 0)$ and $(x_j, y_j, 0)$, respectively. Let the coordinate of the target source be $(x_s, y_s, h)$. The visibility measurement for $P_i$ and $P_j$ is [12]:

$$V_{ij} = \iint_{\xi^2+\eta^2 \leq 1} T_B(\xi, \eta) \frac{F_{ni}(\xi, \eta) F_{nj}^*(\xi, \eta)}{\sqrt{\Omega_i \Omega_j} \sqrt{1 - \xi^2 - \eta^2}} \frac{R_s^2}{L_i L_j} e^{jk(L_j - L_i)} d\xi d\eta \tag{1}$$

where $\xi = \sin\theta \cos\varphi$ and $\eta = \sin\theta \sin\varphi$ are the direction cosines; $(\theta, \varphi)$ are the incident angles of each pixel; $T_B(\xi, \eta)$ stands for the BT of the target source; $F_{ni}(\xi, \eta)$ and $F_{nj}(\xi, \eta)$ are the normalized voltage antenna pattern of the two antennas $P_i$ and $P_j$, and their corresponding antenna equivalent solid angles are $\Omega_i$ and $\Omega_j$; $k$ is the wavenumber ($k = 2\pi/\lambda$ where $\lambda$ is wavelength); $R_s = \sqrt{x_s^2 + y_s^2 + h^2}$ is the distance between the source and the origin of the coordinate; and $L_i$ and $L_j$ are the distances between the antenna elements and the target source. The fringe-washing function has been neglected.

The distance between the antenna element $P_i$ and the origin of the coordinate is $R_i = \sqrt{x_i^2 + y_i^2}$, and the antenna element $P_j$ is $R_j = \sqrt{x_j^2 + y_j^2}$. Then, the distance between the antenna element $P_i$ and the source can be expressed as:

$$\begin{aligned} L_i &= \sqrt{(x_s - x_i)^2 + (y_s - y_i)^2 + h^2} \\ &= \sqrt{R_s^2 + R_i^2 - 2(x_s x_i + y_s y_i)} \\ &= R_s \sqrt{1 + \left(\frac{R_i}{R_s}\right)^2 - \frac{2(x_s x_i + y_s y_i)}{R_s^2}} \end{aligned} \tag{2}$$



A Taylor approximation can be performed, and Equation (2) can be reshaped as:

$$L_i = R_s[1 + \frac{1}{2}(\frac{R_i}{R_s})^2 - \frac{(x_s x_i + y_s y_i)}{R_s^2} + \sigma_i] \tag{3}$$

where $\sigma_i$ is the remainder of the Taylor approximation.

Similarly, the distance between the antenna element $P_j$ and the source can be expressed as: $L_j = R_s[1 + \frac{1}{2}(\frac{R_j}{R_s})^2 - \frac{(x_s x_j + y_s y_j)}{R_s^2} + \sigma_j]$. Therefore, $L_j - L_i$ can be calculated as:

$$L_j - L_i = \frac{R_j^2 - R_i^2}{2R_s} - \frac{x_s(x_j - x_i) + y_s(y_j - y_i)}{R_s} + \sigma_j - \sigma_i \tag{4}$$

In the far-field condition, the approximations, $\frac{R_s^2}{L_i L_j} \approx 1$, $\frac{R_j^2 - R_i^2}{2R_s} \approx 0$, and $\sigma_j - \sigma_i \approx 0$, are applied and Equation (4) can be computed as:

$$\begin{aligned} L_j - L_i \approx \quad & -\frac{x_s(x_j - x_i) + y_s(y_j - y_i)}{R_s} \\ & = -[(x_j - x_i)\xi + (y_j - y_i)\eta] \end{aligned} \tag{5}$$

where $\xi = \frac{x_s}{R_s}$, and $\eta = \frac{y_s}{R_s}$. Thus, the well-known far-field visibility is expressed as

$$V_{ij} = \iint_{\xi^2 + \eta^2 \leq 1} T_B(\xi, \eta) \frac{F_{ni}(\xi, \eta)F_{nj}^*(\xi, \eta)}{\sqrt{\Omega_i \Omega_j}\sqrt{1 - \xi^2 - \eta^2}} e^{-j2\pi(u\xi + v\eta)} d\xi d\eta \tag{6}$$

where $u = \frac{x_j - x_i}{\lambda}$ and $v = \frac{y_j - y_i}{\lambda}$, which are defined as the differences between the positions of the two antenna elements normalized with wavelength. It can be seen from Equation (6) that the relationship between the visibility and the modified BT $T_{\mathrm{mod}}(\xi, \eta) = T_B(\xi, \eta)\frac{F_{ni}(\xi, \eta)F_{nj}^*(\xi, \eta)}{\sqrt{\Omega_i \Omega_j}\sqrt{1 - \xi^2 - \eta^2}}$ becomes a Fourier transform pair. The modified BT can then be recovered:

$$T_{\mathrm{mod}}(\xi, \eta) = \int_{-\infty}^{\infty} \int_{-\infty}^{\infty} V(u, v)e^{j2\pi(u\xi + v\eta)} du dv \tag{7}$$

In an ideal case (equal antenna and neglecting the fringe-washing function), the BT can be obtained by solving the visibility through inverse Fourier transform. In the real case, due to the non-negligible differences in antenna patterns, it is necessary to accurately measure all antenna patterns in the array. The Fourier transform is no longer suitable and the so-called G-matrix method is applied.

For an ASR system, the system input and output can be established. The BT distribution in the target source space is the system input and the visibility is the system output. Ideally, the relation between visibility V and BT $T_B$ can be expressed as

$$V(u, v) = \int \int g(u, v, \xi, \eta) T_B(\xi, \eta) d\xi d\eta \tag{8}$$

where $g(u, v, \xi, \eta)$ is the system impulse response.

The target source can be regarded as the sum of a finite number of discrete sampling points, and the integral of the above Equation (8) can be written in the following discrete form:

$$\underset{(M\times1)}{\mathbf{V}} = \underset{(M\times P)}{\mathbf{G}} \underset{(P\times1)}{\mathbf{T}_B}$$

$$\begin{bmatrix} V_1 \\ V_2 \\ \cdots \\ V_M \end{bmatrix} = \begin{bmatrix} g_1(\xi_1, \eta_1) & g_1(\xi_2, \eta_2) & \cdots & g_1(\xi_P, \eta_p) \\ g_2(\xi_1, \eta_1) & g_2(\xi_2, \eta_2) & \cdots & g_2(\xi_P, \eta_p) \\ \cdots & \cdots & \cdots & \cdots \\ g_M(\xi_1, \eta_1) & g_M(\xi_2, \eta_2) & \cdots & g_M(\xi_P, \eta_p) \end{bmatrix} \cdot \begin{bmatrix} T_B(\xi_1, \eta_1) \\ T_B(\xi_2, \eta_2) \\ \cdots \\ T_B(\xi_P, \eta_p) \end{bmatrix} \tag{9}$$

where **V** represents the visibility vector; **G** represents the impulse response matrix of the system; and $\mathbf{T}_B$ represents the BT vector. *M* is the number of visibility and *P* is the number of pixels of BT. Equation (9) is a linear system of equations [17]. To satisfy Shannon's sampling theorem and ensure the accuracy of image inversion, *P* needs to be sufficiently large, usually $P > 3\,M$ [18]. As a consequence, Equation (9) is an under-constrained problem, and the inverse problem is ill-posed. There are multiple solutions to solve BT, such as the Moore–Penrose Pseudo inverse, the singular value decomposition (SVD), or the conjugate–gradient method [19]. The G-matrix can be obtained by system measurement, but the measurement of the G-matrix will be affected by measurement errors and system noise, and the measurement of the G-matrix for a large two-dimensional antenna array requires a large amount of work. Therefore, the G-matrix is normally calculated from the ground-based characterization data of the ASR. Each element in the G-matrix is composed of auxiliary data such as antenna pattern and antenna position, which can be written as:

$$g_{ij}(\xi,\eta) = \frac{F_{ni}(\xi,\eta)F_{nj}^*(\xi,\eta)}{\sqrt{\Omega_i\Omega_j}\sqrt{1-\xi^2-\eta^2}}e^{-j2\pi(u\xi+v\eta)}\Delta\xi\Delta\eta \tag{10}$$

In far-field conditions, the reconstructed BT can be obtained by solving the G-matrix and the visibility.

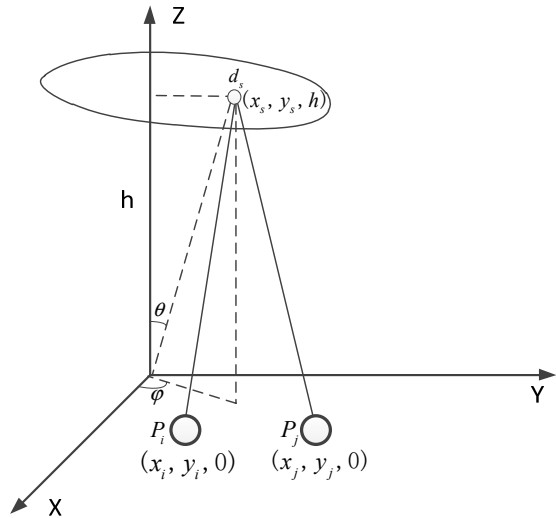

**Figure 1.** Illustration of ASR imaging.

## 3. Near-Field Imaging Based on the Near-Field Distance

### 3.1. Near-Field G-Matrix

When the ASR is tested and applied on the ground, the target source is in the near-field condition due to size limitations. The visibility no longer satisfies Equation (6), and the BT cannot be correctly reconstructed using the far-field imaging method. Based on the derivation of Equations (1)–(4), the visibility in near-field conditions can be expressed as

$$V_{ij} = \iint\limits_{\xi^2+\eta^2\leq 1} T_{\mathrm{mod}}(\xi,\eta)\frac{R_s^2}{L_iL_j}e^{\frac{R_j^2-R_i^2}{2R_s}-j2\pi(u\xi+v\eta)+(\sigma_j-\sigma_i)}d\xi d\eta \tag{11}$$

$\frac{R_j^2-R_i^2}{2R_s}$ and $\sigma_j - \sigma_i$ are the phase error terms introduced in the near field. $\frac{R_s^2}{L_iL_j}$ is the amplitude error term. Due to the effect of these near-field errors, the visibility and the

modified BT no longer satisfy the Fourier transform. When the remainder of the Taylor approximation $\sigma_j - \sigma_i$ is ignored, the visibility with near-field error can be written as

$$V_{ij} = \iint\limits_{\xi^2 + \eta^2 \leq 1} T_{\text{mod}}(\xi, \eta) \frac{R_s^2}{L_i L_j} e^{-j2\pi(u\xi + v\eta)} e^{\frac{R_j^2 - R_i^2}{2R_s}} d\xi d\eta \tag{12}$$

The coordinate system is established on the array plane, and the pixels are divided in the FOV observed using the radiometer. If the distance h between the array plane and the target source plane is known, the coordinates of each pixel in the FOV can be calculated as

$$(x_s, y_s, h) == (\frac{\xi_s h}{\sqrt{1 - \xi_s^2 - \eta_s^2}}, \frac{\eta_s h}{\sqrt{1 - \xi_s^2 - \eta_s^2}}, h) \tag{13}$$

Then, $R_s$ can be calculated as:

$$R_s = \sqrt{x_s^2 + y_s^2 + h^2} = \frac{h}{\sqrt{1 - \xi_s^2 - \eta_s^2}} \tag{14}$$

$R_i$ and $R_j$ can be calculated from the coordinate positions of the antennas. For any antenna pair in the array, the near-field error term $\frac{R_j^2 - R_i^2}{2R_s}$ can be calculated.

The distance between the antenna element $P_i$ in the array and each pixel in the FOV can be obtained as follows:

$$\begin{aligned} L_i &= \sqrt{(x_s - x_i)^2 + (y_s - y_i)^2 + h^2} \\ &= \sqrt{(\frac{\xi_s h}{\sqrt{1 - \xi_s^2 - \eta_s^2}} - x_i)^2 + (\frac{\eta_s h}{\sqrt{1 - \xi_s^2 - \eta_s^2}} - y_i)^2 + h^2} \end{aligned} \tag{15}$$

Similarly, $L_j$ can be obtained by replacing $(x_i, y_i, 0)$ in the above equation with $(x_j, y_j, 0)$ of the antenna element $P_j$. Then, $\frac{R_s^2}{L_i L_j}$ can be calculated.

By comparing Equation (6) and Equation (12), the near-field error information is added to the G-matrix, and updating Equation (10), the near-field G-matrix $\mathbf{G}^{NF}$ is obtained. Each element in the near-field G-matrix can be written as:

$$\begin{aligned} g_{ij}^{NF}(\xi, \eta) &= g_{ij}(\xi, \eta) * NF \\ &= \frac{F_{ni}(\xi, \eta) F_{nj}^*(\xi, \eta)}{\sqrt{\Omega_i \Omega_j} \sqrt{1 - \xi^2 - \eta^2}} \frac{R_s^2}{L_i L_j} e^{-j2\pi(u\xi + v\eta)} e^{\frac{R_j^2 - R_i^2}{2R_s}} \Delta\xi \Delta\eta \end{aligned} \tag{16}$$

The near-field error term in the visibility can be corrected by the updated near-field G-matrix $\mathbf{G}^{NF}$. The error brought by the near-field is compensated to the G-matrix by using the distance h. When reconstructing BT, the near-field error in the visibility can be corrected by using the near-field G-matrix $\mathbf{G}^{NF}$. The BT image can be obtained, which is the same as the far-field one.

### 3.2. F-Matrix

When the term $\sigma_j - \sigma_i$ is not negligible, it needs to be considered in the near-field BT reconstruction. Since this part cannot be computed simply and does not compensate the G-matrix, as in the previously proposed method, a new solution needs to be proposed. Based on the visibility Equation (1), the relationship between visibility V and BT $T_B$ can be expressed as a matrix–vector multiplication, hereafter referred to as the F-matrix:

$$\underset{(M \times 1)}{\mathbf{V}} = \underset{(M \times P)}{\mathbf{F}} \underset{(P \times 1)}{\mathbf{T}_B} \tag{17}$$

Each element of the F-matrix can be expressed as

$$f_{ij}(\xi,\eta) = \frac{F_{ni}(\xi,\eta)F_{nj}^*(\xi,\eta)}{\sqrt{\Omega_i \Omega_j}\sqrt{1-\xi^2-\eta^2}}\frac{R_s^2}{L_i L_j}e^{jk(L_j-L_i)}\Delta\xi\Delta\eta \tag{18}$$

The elements in the F-matrix are similar to those in the G-matrix, with the main difference being that the phases of the two are different. The phase difference between the target source and each element of the array in the F-matrix is the actual path phase difference, and the far-field approximation is not used. Therefore, the F-matrix can be applied to near-field imaging as long as the path between the target source and each element of the array can be calculated. Like the near-field G-matrix presented above, the F-matrix also requires knowledge of the near-field distance to enable near-field BT reconstruction.

From the above derivation, the near-field G-matrix and the F-matrix depend on antenna pattern, antenna position, and near-field distance.

The two near-field imaging methods described above can achieve BT reconstruction for both point sources and extended sources. The imaging effect will be verified by simulations and experiments in the following section.

**4. Simulation**

In order to verify the two near-field imaging methods, simulations were performed for imaging point sources and extended sources. A 10-element Y-array was established for simulation verification, which is consistent with the aperture synthesis system configuration used for the experiments. The array consists of three elements in one arm and one element in the center. The minimum spacing between elements is 0.88 $\lambda$, and the $\lambda$ is 0.212 m. The array arrangement is shown in Figure 2. According to the equation of far field $2\frac{D^2}{\lambda}$ and $20\frac{D^2}{\lambda}$ (D is the array size), the general far-field distance of this array is about 5 m, and the absolute far-field distance is about 50 m.

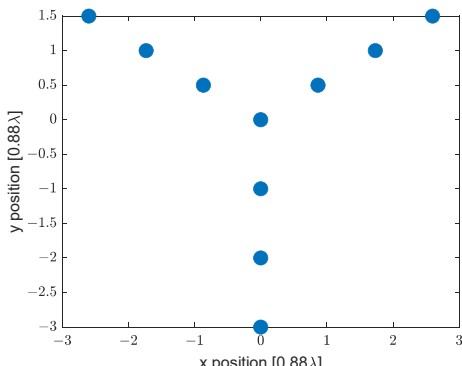

**Figure 2.** Array arrangement.

Simulations are carried out in far-field conditions and near-field conditions, respectively. Near-field visibilities are estimated according to Equation (1) and far-field visibilities according to Equation (6). In the simulation, a Blackman window is applied to smooth ripples and sidelobes levels in the images. In addition, all antenna patterns are set to be identical, decorrelation effects are negligible, and systematic errors are not taken into account.

*4.1. Point Source*

Figure 3 shows the imaging simulation results of the point source placed at boresight when the far-field imaging method (G-matrix method) is applied. It can be seen from Figure 3a that the reconstructed far-field point source shows good imaging quality. In the near-field simulations, the distance between the array and the target source is set to 2 m, 5 m, and 50 m, respectively. Imaging results at different near-field distances are

shown in Figure 3b–d. Obviously, the far-field imaging method brings large distortions to the near-field point source. The distorted shape of the point source is similar to the shape of the array in the Y-shape. Moreover, the distortion becomes more severe as the distance gets closer. The near-field distortion is present at a general far-field distance of 5 m, and the distortion essentially disappears at an absolute far-field distance of 50 m. The imaging result at a distance of 50 m is in good agreement with that obtained in the far-field condition.

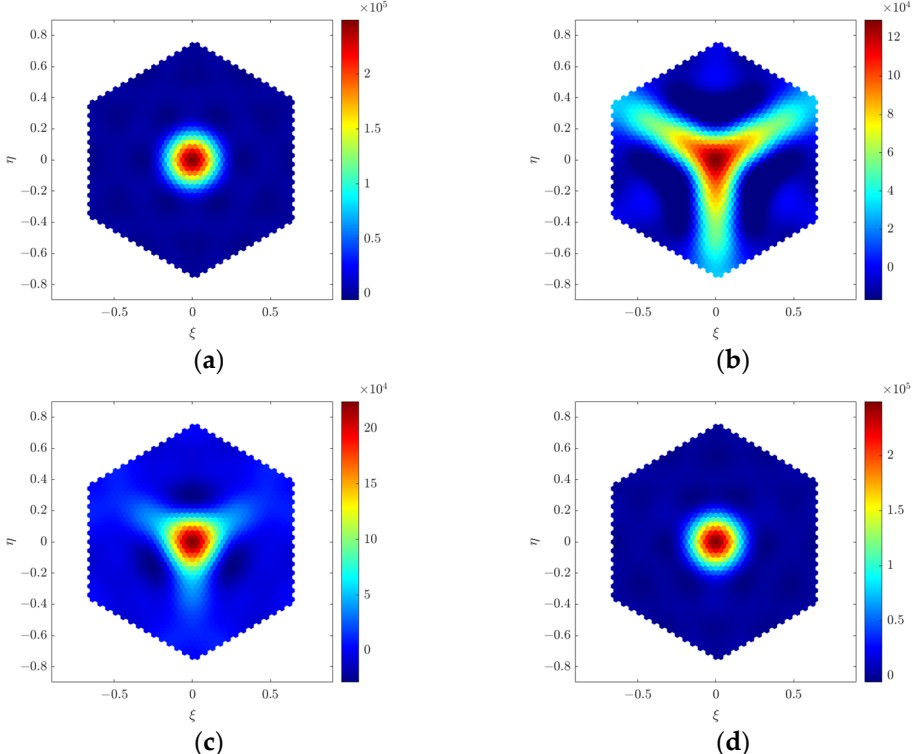

**Figure 3.** Imaging simulation results of a point source. (**a**) Far-field point source. (**b**) Near-field point source at a distance of 2 m. (**c**) Near-field point source at a distance of 5 m. (**d**) Near-field point source at a distance of 50 m.

In order to compare with the experimental results, a near-field point source at a distance of 2.46 m is simulated. The two near-field imaging methods (near-field G-matrix and F-matrix) proposed in this paper are applied. Figure 4 shows the imaging simulation results of the point source placed in the center of the FOV. Figure 5 shows the imaging simulation results of the point source placed off the center of the FOV. After using the two near-field imaging methods, the point source shape becomes concentrated from the heavily distorted Y-shape. It is shown that both the near-field G-matrix and the F-matrix based on the near-field distance can effectively correct the distortion caused by the near-field errors. The proposed imaging methods allow the near-field point source to obtain the images as if the source were in the far-field conditions. Additionally, these two methods have no restrictions on the position of the target in the FOV. Both methods can obtain high-quality images regardless of whether the target is located in the center of the FOV.

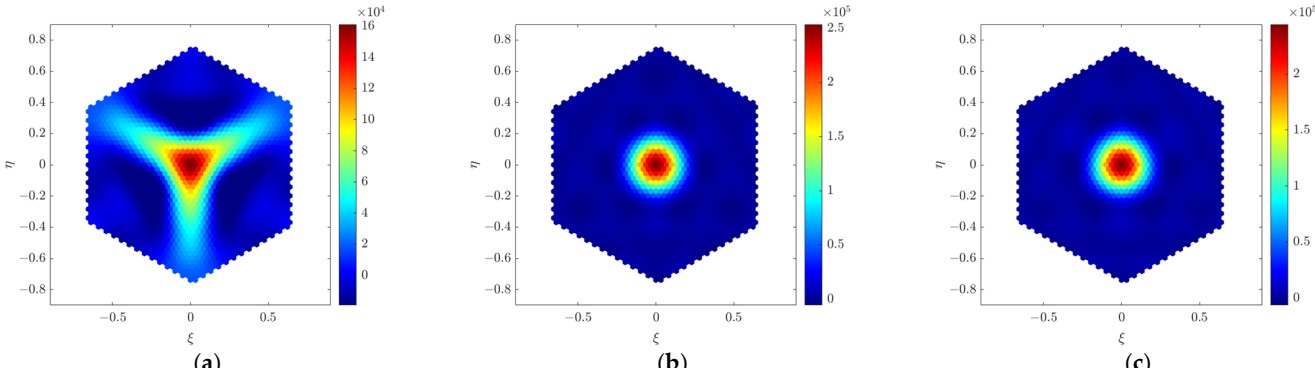

**Figure 4.** Imaging simulation results of a near-field point source placed in the center of the FOV (**a**) G-matrix method. (**b**) Near-field G-matrix method. (**c**) F-matrix method.

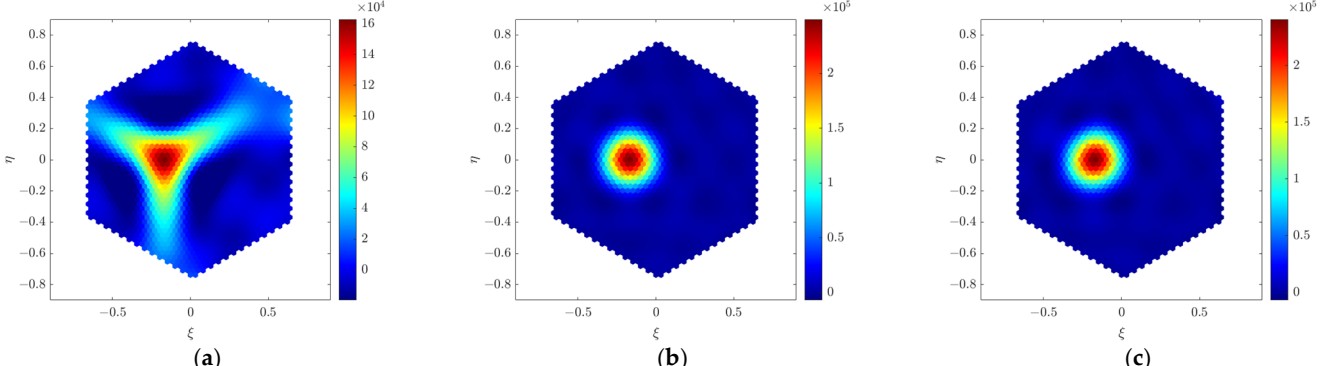

**Figure 5.** Imaging simulation results of a near-field point source placed off the center of the FOV (**a**) G-matrix method. (**b**) Near-field G-matrix method. (**c**) F-matrix method.

### 4.2. Extended Source

The extended source is set to a rectangular scene with a side length of 0.4 at $T_B = 200K$. Figure 6a shows the BT distribution. Figure 6b shows the imaging result of the extended source in the far-field condition. In the near-field simulations, the distance between the extended source and the array is set to 2.46 m. The RMSE difference $\Delta T$ between near-field and far-field images is computed. The calculation of $\Delta T$ is as follows:

$$\Delta T = \sqrt{\frac{1}{m}\sum_{k=1}^{m}\left[T_{NF}(\xi_k,\eta_k) - T_{FF}(\xi_k,\eta_k)\right]^2} \tag{19}$$

where $m$ is the number of pixels. $T_{NF}(\xi_k,\eta_k)$ is the reconstructed BT of pixel $(\xi_k,\eta_k)$ under near-field conditions. $T_{FF}(\xi_k,\eta_k)$ is under far-field conditions.

The results of an imaging simulation of the near-field extended source at a distance of 2.46 m are shown in Figure 7. The error of the G-matrix method is 32.2 K, which is much different from that of the far-field imaging results. The shape of the rectangular extended source is completely distorted and is extended in a Y-shape. The error of the near-field G-matrix method is reduced to 5.1 K and the error of the F-matrix is further reduced to 3 K. The results of the two near-field imaging methods are similar in shape, and they are close to the shape of the far-field imaging, especially the F-matrix method.

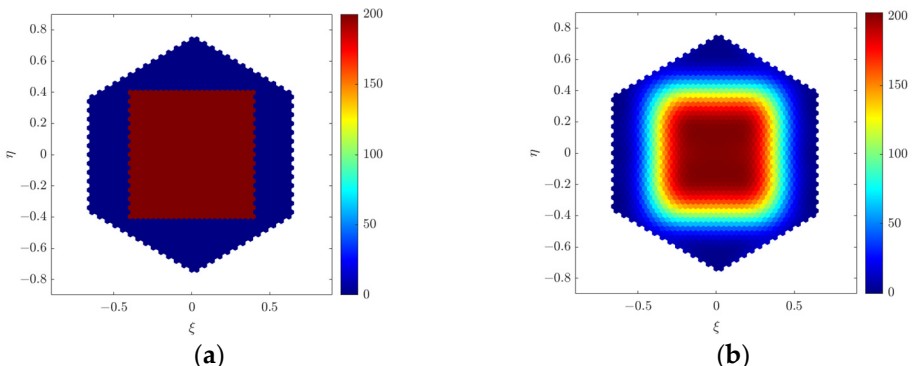

**Figure 6.** (**a**) BT distribution of extended source. (**b**) Far-field imaging result.

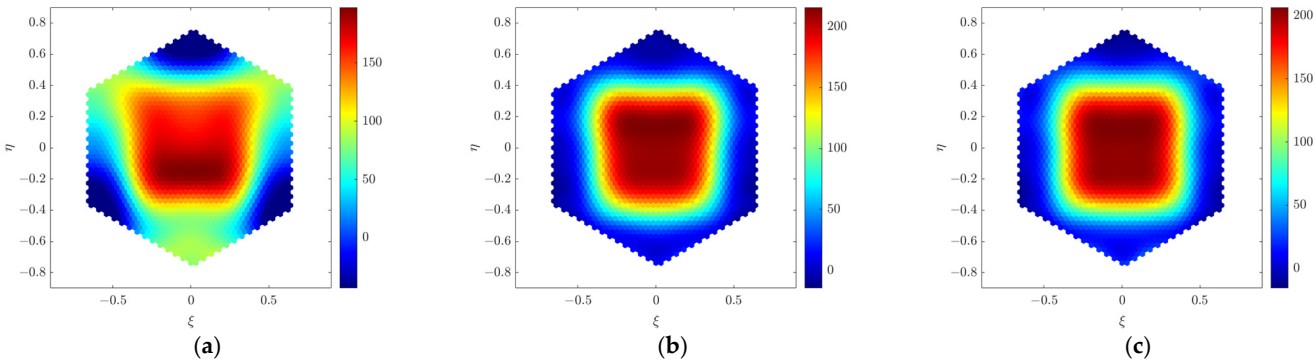

**Figure 7.** Imaging simulation results of the near-field extended source at a distance of 2.46 m. (**a**) G-matrix method, $\Delta T$ = 32.2 K. (**b**) Near-field G-matrix method, $\Delta T$ = 5.1 K. (**c**) F-matrix method, $\Delta T$ = 3 K.

Simulation results indicate that both near-field imaging methods can effectively reduce near-field errors. Moreover, the F-matrix method gives better results than the near-field G-matrix due to its smaller error. This indicates that the remainder of the Taylor approximation has an impact on the near-field imaging, which is the reason why the near-field G-matrix method has a larger error than the F-matrix method.

## 5. Experiment

Indoor experiments were carried out in an anechoic chamber. Figure 8a shows the system used in the experiments. It is an L-band 10-element Y-array ASR. The antenna spacing is 0.88λ. A rectangle horn antenna is used to generate the point source signal as shown in Figure 8b. The distance between the transmitting antenna and the antenna array of the radiometer is 10.06 m and 2.46 m, respectively. Figure 9 shows the photographs of the experimental scene for imaging the near-field point source. As mentioned in the previous section, the general far field and the absolute far field are 5 m and 50 m. Therefore, the observed target at a distance of 2.46 m is in the near-field area of the system. The observed target at a distance of 10.06 m is in the general far field of the system.

Figures 10a, 11a and 12a are the indoor imaging results of point sources with the G-matrix method. Both images are distorted; in particular the point source at a distance of 2.46 m follows a Y-shape. With the use of near-field imaging methods based on near-field distance, the image quality is greatly improved. The near-field G-matrix is used in Figures 10b, 11b and 12b, while the F-matrix is used in Figures 10c, 11c and 12c. From the point source imaging results, both methods provide good correction of near-field errors. In Figures 10 and 12, the point source is not placed in the center of the FOV, which verifies the imaging in the off-axis region. Moreover, the experimental results (see Figures 11 and 12) are consistent with the simulation results (see Figures 4 and 5).

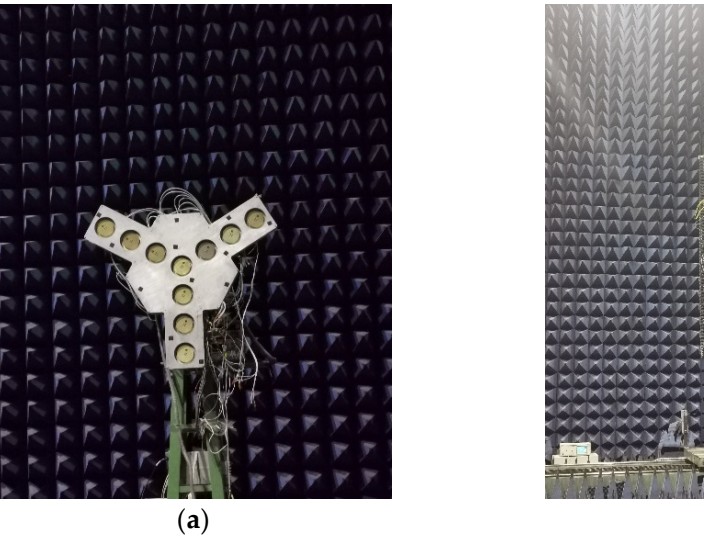

**Figure 8.** The transmitting and receiving devices of the experiment. (**a**) ASR. (**b**) Transmitting horn antenna.

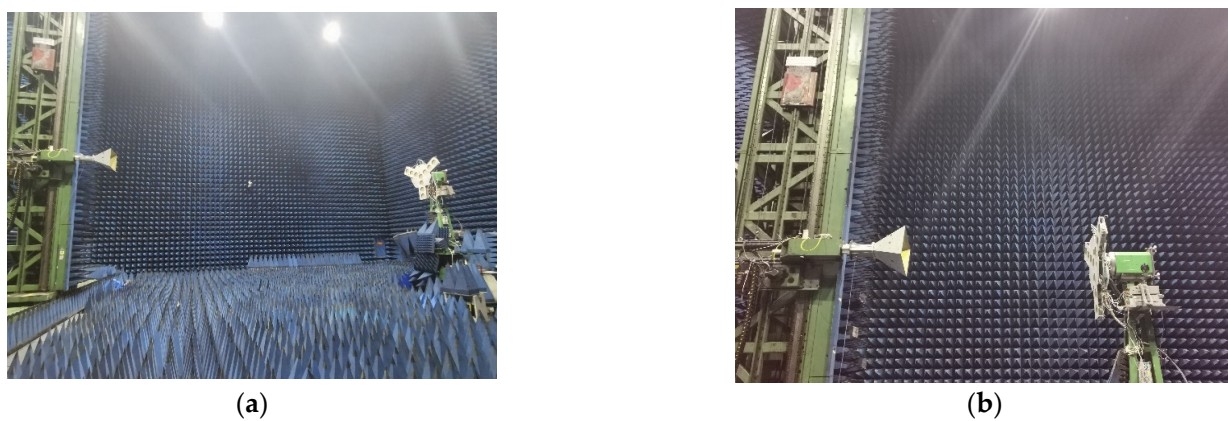

**Figure 9.** The experimental scene. (**a**) 10.06 m. (**b**) 2.46 m.

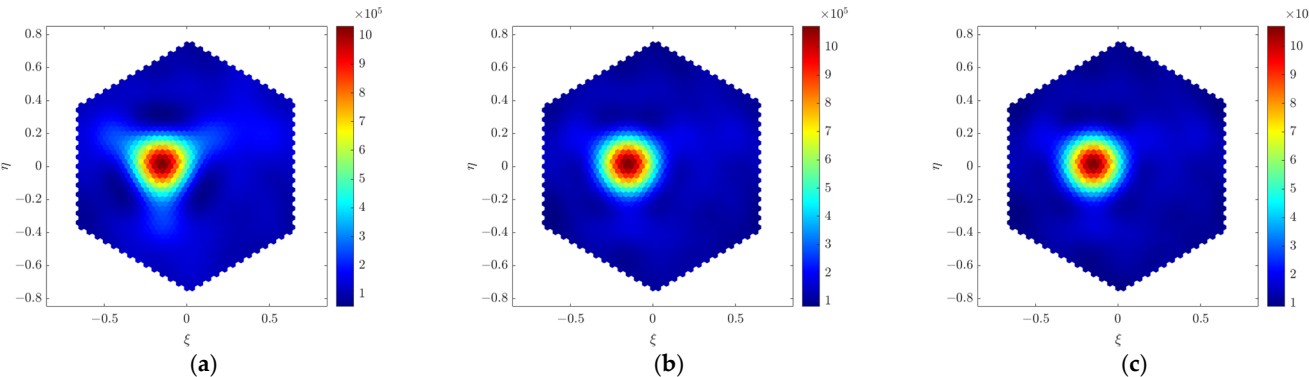

**Figure 10.** Imaging results of a point source at the experimental distance of 10.06 m. (**a**) G-matrix method. (**b**) Near-field G-matrix method. (**c**) F-matrix method.

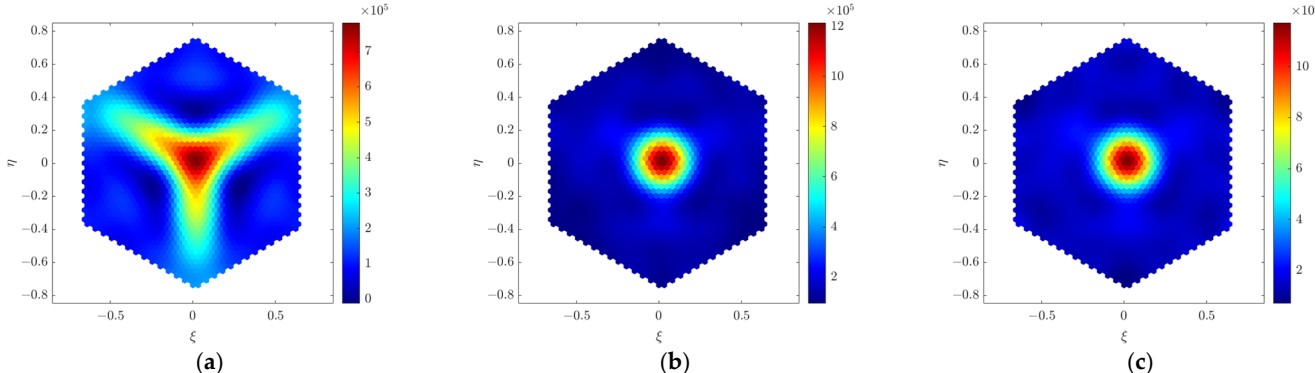

**Figure 11.** Imaging results of a point source placed in the center of the FOV at the experimental distance of 2.46 m. (**a**) G-matrix method. (**b**) Near-field G-matrix method. (**c**) F-matrix method.

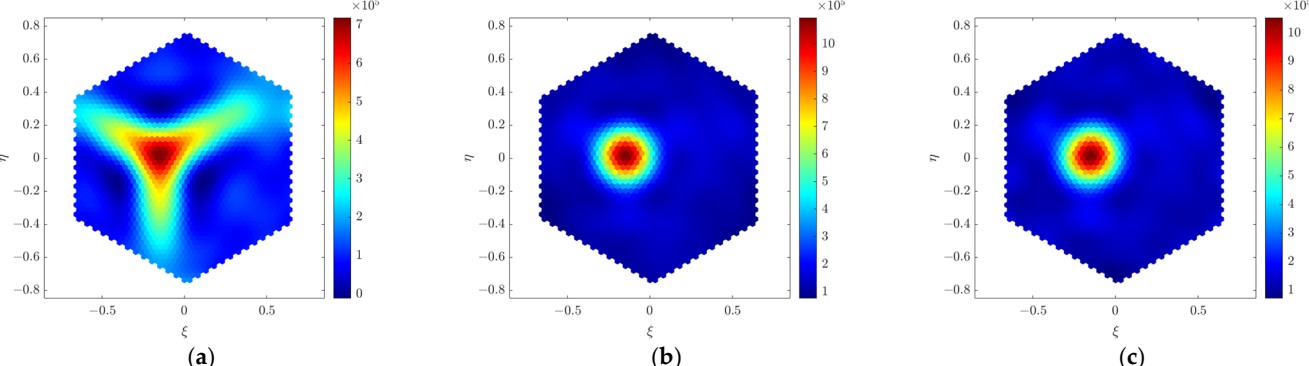

**Figure 12.** Imaging results of a point source placed off the center of the FOV at the experimental distance of 2.46 m. (**a**) G-matrix method. (**b**) Near-field G-matrix method. (**c**) F-matrix method.

Simulations and experiments were performed to measure the spatial resolution. For rectangular windows, the spatial resolution can be approximately computed from:

$$\Delta\theta_{3\text{dB}} \approx \frac{\pi/2}{\Delta u_{\max}} \tag{20}$$

where $\Delta u_{\max} = 2\sqrt{3}Nd$ is the maximum dimension of the Y-array, N = 3 is the number of antenna elements in each arm, and d = $0.88\lambda$ is the spacing of the array. According to Equation (20), the spatial resolution is $9.84°$ approximately.

The spatial resolution was measured from the imaging results of the point source at a distance of 2.46 m, and the point source is aligned with the center of the array. A rectangular window was applied to the visibilities. Figure 13 shows the profile plot for a single point at $\eta = 0$. Figure 13a is the simulation result in the far field, and the resolution is $10.1°$. Figure 13b,c are the experimental results in the near field. The reconstructed images using the near-field G-matrix and F-matrix methods are consistent. The corresponding spatial resolution is $10.3°$ and $10.5°$. Theoretical analysis, simulation, and experimental results are matched.

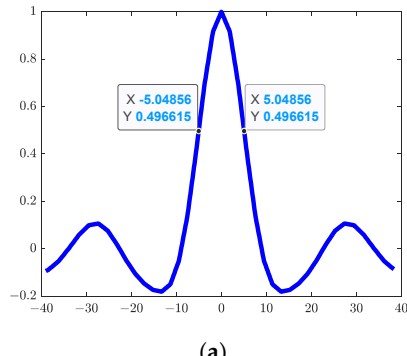 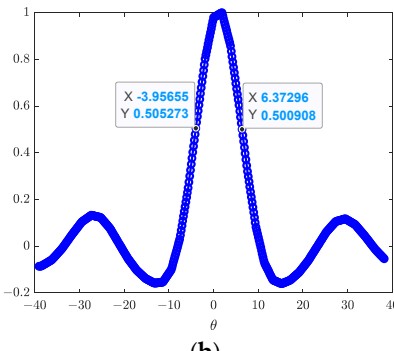 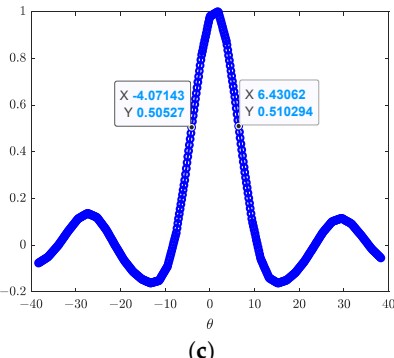

(**a**)  (**b**)  (**c**)

**Figure 13.** Spatial resolution. (**a**) Simulation result. (**b**) Experimental result by using the near-field G-matrix method. (**c**) Experimental result by using the F-matrix method.

## 6. Conclusions

Near-field imaging is useful and important for large ASRs. Near-field imaging enables ASRs to perform performance evaluations such as system imaging under limited near-field conditions. In addition, there is also a demand for the near-field imaging of ASRs in security and other fields. In the near-field condition, the G-matrix imaging method introduces near-field errors that distort the shape of the imaging target. This paper explored the relationship between visibility and BT under near-field conditions and proposed two near-field imaging methods based on the near-field distance: the near-field G-matrix and the F-matrix. The novelty of these methods is that they only need to generate improved resolving matrices, and the rest of the processing process is the same as the traditional brightness temperature reconstruction process for ASRs. In addition, they can be used not only for point sources but also for extended sources, and there are no requirements for the location of the target.

Simulations and experimental results demonstrate that both methods can be used for near-field imaging. In addition, both methods are suitable for near-field imaging of point sources and extended sources. According to the imaging simulation results of the near-field extended source, the imaging error using the far-field G-matrix is 32.2 K. After applying the near-field G-matrix and F-matrix methods, the errors are reduced to 5.1 K and 3 K, respectively. The results demonstrate the effectiveness of both methods and show that the F-matrix method provides a better correction to the near-field error. The imaging experimental results of a point source verify that these two near-field imaging methods can achieve near-field imaging in the off-axis region.

**Author Contributions:** Conceptualization, Y.W. and Y.L.; methodology, Y.W., G.S. and H.D.; software, Y.W., X.Y. and H.D.; validation, Y.W., D.W., P.L. and G.S.; investigation, R.L., X.Y. and H.L.; resources, R.L. and H.L.; writing—original draft preparation, Y.W., Y.L. and G.S.; writing—review and editing, Y.W. and Y.L.; visualization, Y.W. and G.S.; supervision, P.L. and D.W. All authors have read and agreed to the published version of the manuscript.

**Funding:** This research was funded by the Pre-research Project of Civil Aerospace Technology of China (D040202).

**Data Availability Statement:** The data presented in this study are available on request from the corresponding author. The data are not publicly available due to privacy reasons.

**Conflicts of Interest:** The authors declare no conflicts of interest.

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
