# Peer review of "A Near-Field Imaging Method Based on the Near-Field Distance for an Aperture Synthesis Radiometer"

_remotesensing, doi:10.3390/rs16050767_

Round 1

Reviewer 1 Report

Comments and Suggestions for Authors

This paper deals with passive imaging radiometry by aperture synthesis in the near field region of the antenna array. This is a very interesting subject however an error, or an assumption, made by the authors can be read in the very first equation.

The original relation between the complex visibility sample for a given pair of elementary antennas and the brightness temperature distribution of the scene observed by the antenna array is found in two very well-known books which are not cited by the authors:

M. Born and E. Wolf , “Principles of Optics”, 7th ed., Cambridge University Press, 1999.

A.R. Thompson, J.W. Moran and G.W. Swenson, “Interferometry and Synthesis in Radio Astronomy”, 3rd ed., Springer International Publishing, 2017.

When deriving their equation (1), the authors should start from equation (20) page 573 in “Principles of Optics” or from equation (15.6) page 770 in “Interferometry and Synthesis in Radio Astronomy” as well, so that they will not forgot to divide the exponential term by the distances Li and Lj (these distances are named R1 and R2 in those two books) which is there to account for the very well-known decreasing of the spherical waves with the square of the distance. If they proceed like this, the authors will start their paper from the correct equation which is nothing but equation (1) in their reference [12] or [14] (where these two distances are named rk and rj).

If the authors decide to neglect this term, more exactly the ratio between the square of r and the product between rk and rj, they should clearly explain why, what are the underlying assumptions and check if they are compatible with near field imaging.

Comments on the Quality of English Language

As far as this reviewer can estimate it, the quality of english is at the level expected for MDPI publishing.

Author Response

Thank you for pointing this out. We revised the mistake according to your comment. See the attachment for details.

Reviewer 2 Report

Comments and Suggestions for Authors

The authors have presented their research well. I have a few minor concerns to be addressed by the author to make the manuscript complete in all aspects.

Please add a suitable reference to Eq. 1.

Is there any reason why the experiment is conducted with an offset point source while simulations are done at the centre? It will be interesting to see how well the experiment compares with simulations with and without offset. Further, no experiment is conducted for the extended source. It will be interesting to see the error values obtained through experiments to check the consistency of simulations for various source types.

Author Response

Thank you for your careful reading and good comments on our manuscript. To improve the manuscript, we have done further work and revised our manuscript according to your comments. See the attachment for details.

Reviewer 3 Report

Comments and Suggestions for Authors

The authors describe a near field imaging method applicable to synthetic aperture radars.  I would have several comments regarding this manuscript, as shown below.

1.      Several manuscripts have been published related to the subject of near-field imaging method, 1) beam forming technique, near field to far field transformation, etc. The authors group has also published several manuscripts in this subject. The superiority of this manuscript is not clear including the simulation and experimental results. The authors should clearly discuss the merits of the present method in more detail.

2.      There could be several issues related to near-field experiment, such as, multiple reflections between the subjects and spurious reflections. Please discuss whether these issues affect the present analysis.  

3.      I understand that the study of near-field imaging is useful for especially passive imaging. However, it may not clear to me for SAR radar case. For example, if we use X-band radar, the distance between the subject and receiver would be 1-2 m at most. We do not need all-weather system. The authors could discuss this point more clearly. 

4.      Minor point: 

1)      Please indicate the sources (references) for Equation (1), if any.

2)      Please describe the horizontal and vertical axes. 

Author Response

(The authors gave the same response as above.)

Round 2

Reviewer 3 Report

Comments and Suggestions for Authors

I think, the authors have managed to address most of reviewer’s comments. This manuscript warrants publication in Remote Sensing.

One small question is related to comment 2.

I understand that the experiment is performed in the anechoic chambers, however, there could be multi-reflection between transmitting antenna and receiver antenna in near field condition.. Also in the actual condition there could be several spurious reflections, Does your analysis method cause a problem under these conditions ?   

Author Response

Thank you for your careful reading and good comments on our manuscript. See the attachment for details.
